# Charge state-dependent symmetry breaking of atomic defects in transition metal dichalcogenides

Feifei Xiang [1,8], Lysander Huberich[1,8], Preston A. Vargas [2,8], Riccardo Torsi[3], Jonas Allerbeck [1], Anne Marie Z. Tan[2,4], Chengye Dong[5], Pascal Ruffieux [1], Roman Fasel [1], Oliver Gröning[1], Yu-Chuan Lin[3,6], Richard G. Hennig [2], Joshua A. Robinson[3,5,7] & Bruno Schuler [1] ✉

The functionality of atomic quantum emitters is intrinsically linked to their host lattice coordination. Structural distortions that spontaneously break the lattice symmetry strongly impact their optical emission properties and spin-photon interface. Here we report on the direct imaging of charge state-dependent symmetry breaking of two prototypical atomic quantum emitters in mono- and bilayer $MoS_2$ by scanning tunneling microscopy (STM) and non-contact atomic force microscopy (nc-AFM). By changing the built-in substrate chemical potential, different charge states of sulfur vacancies ($Vac_S$) and substitutional rhenium dopants ($Re_{Mo}$) can be stabilized. $Vac_S^{-1}$ as well as $Re_{Mo}^0$ and $Re_{Mo}^{-1}$ exhibit local lattice distortions and symmetry-broken defect orbitals attributed to a Jahn-Teller effect (JTE) and pseudo-JTE, respectively. By mapping the electronic and geometric structure of single point defects, we disentangle the effects of spatial averaging, charge multistability, configurational dynamics, and external perturbations that often mask the presence of local symmetry breaking.

Mastering the controlled introduction of defects and impurities in semiconductors has proven pivotal to their transformative technological success. As miniaturization reaches atomic length scales and quantum applications emerge, stochastic distributions make way for single impurity engineering[1,2]. This need for precise defect manipulation extends to the realm of artificial atom qubits, a cornerstone of quantum sensing and quantum communication[3,4]. Atomic quantum emitters based on defects in solids, rely on spin-selective optical decay pathways enabling high-fidelity spin initialization and readout even at room temperature[5]. Two-dimensional (2D) materials such as

hexagonal boron nitride or transition metal dichalcogenides (TMDs) emerged as a game-changing platform to host such quantum emitters because they lack surface dangling bonds that degrade spin coherence or photon distinguishability[6,7]. Furthermore, 2D materials exhibit increased extraction efficiency[8], integrate seamlessly with quantum optical devices[9,10], and support electrical fine-tuning[11,12] and spatial engineering[10,13] of defects and their associated electronic states. Regardless of the host material, the local defect symmetry governs the level structure and optical selection rules that ultimately provide the spin-photon interface. Spontaneous symmetry breaking from Jahn-

[1]nanotech@surfaces Laboratory, Empa – Swiss Federal Laboratories for Materials Science and Technology, Dübendorf 8600, Switzerland. [2]Department of Materials Science and Engineering, University of Florida, Gainesville, FL 32611, USA. [3]Department of Materials Science and Engineering, The Pennsylvania State University, University Park, PA 16082, USA. [4]Institute of High Performance Computing (IHPC), Agency for Science, Technology and Research (A*STAR), Singapore 138632, Republic of Singapore. [5]Two-Dimensional Crystal Consortium, The Pennsylvania State University, University Park, PA 16802, USA. [6]Department of Materials Science and Engineering, National Yang Ming Chiao Tung University, Hsinchu City 300, Taiwan, ROC. [7]Department of Chemistry and Department of Physics, The Pennsylvania State University, University Park, PA 16802, USA. [8]These authors contributed equally: Feifei Xiang, Lysander Huberich, Preston A. Vargas. ✉e-mail: bruno.schuler@empa.ch

Teller distortions and strain fields can therefore dramatically change the radiative recombination rates, emission wavelength and polarization[14,15]. Jahn-Teller systems, in particular, can strongly influence spin-orbit coupling (SOC) that gives rise to the inter-system crossing, which yields the spin contrast[15–17]. Therefore, understanding the mechanisms that break the coordination symmetry of single atom qubits as a function of their charge state and external factors such as strain are decisive to tailor their functionality.

Although it is well known that TMDs and other transition metal compounds exhibit enhanced susceptibility towards Jahn-Teller effect (JTE), detecting JTE experimentally is challenging, because configurational dynamics and spatial averaging can obscure the minute local distortions[18]. Additionally, these distortions are often affected by external perturbations, which can complicate the interpretation of experimental results[18]. Therefore, even though the JTE and its extensions[19] have been discovered almost a century ago, its abundance and significance in different materials systems has only gradually been revealed and is still mostly inferred theoretically by ab initio methods[18,20]. While a number of TMD point defects including chalcogen vacancies[21,22] and substitutional transition metal dopants[23] have been predicted to undergo spontaneous symmetry breaking, it has only rarely been verified experimentally using aberration-corrected transmission electron microscopy[24], and no direct experimental observation of symmetry-broken electronic states has been reported to date.

Here, we present direct experimental and theoretical evidence for a Jahn-Teller driven symmetry breaking of chalcogen vacancies[13] and rhenium-based quantum emitters[25] in MoS$_2$. Scanning tunneling microscopy (STM) orbital imaging[26], CO-tip noncontact atomic force microscopy (nc-AFM) measurements[27], and density functional theory (DFT) reveal the symmetry-broken defect orbitals and the structural distortion for negatively charged sulfur vacancy (Vac$_S^{-1}$) and rhenium substituting for molybdenum in its neutral and negative charge state (Re$_{Mo}^0$, Re$_{Mo}^{-1}$). We reveal that Vac$_S^{-1}$ coexist in both a symmetric and distorted geometry, which may explain why the most studied TMD

defect has evaded the spectroscopic detection of spontaneous symmetry breaking so far. Charge state tristability of Re dopants is achieved by varying the chemical potential via a different work function of the substrate. We find larger domains where the structural distortions are aligned, which may indicate a non-negligible strain field in the epitaxially-grown MoS$_2$ samples. Surprisingly, the comparatively broad defect resonances of some Re impurities exhibit a continuous transformation of the measured defect orbital, suggesting a configurational continuum as a consequence of the flat potential energy surface experienced by the Re impurity.

## Results and Discussion

### Jahn-Teller Driven Symmetry-Breaking of Vac$_S^{-1}$ in MoS$_2$

Chalcogen vacancies are the most prevalently discussed point defect in TMDs due to their low formation energy[21], frequent detection in transmission electron microscopy[28,29], and attributed sub-band gap emission in optical spectroscopy[30,31]. Despite their disputed presence at ambient conditions due to their high reactivity[32], recent strategies involving deliberate vacancy generation by annealing or ion bombardment and protection by vacuum or inert capping layers made it possible to directly measure unpassivated vacancies in TMDs[13,31,33].

Figure 1a shows a STM image of monolayer MoS$_2$ grown by metal-organic chemical vapor deposition (MOCVD) on quasi-freestanding epitaxial graphene on 6H-silicon carbide (0001) (QFEG). S vacancies (Vac$_S$) were induced by annealing the sample up to 900 °C in ultrahigh-vacuum[33]. Apart from common (unintentional) as-grown defects like O$_S$, CH$_S$, Cr$_{Mo}$, and W$_{Mo}$[34] (see Supplementary Fig. 2) that are also present at much lower annealing temperatures, we find the anticipated Vac$_S$ in both the upper and lower S layer as previously identified[33,35]. All S vacancies are negatively charged because the QFEG substrate sets the Fermi level relatively high up in the MoS$_2$ band gap, such that the lowest Vac$_S$ in-gap state becomes occupied. The negative charge can be inferred from the band bending-induced dark halo around the defect at positive sample bias[36] or by Kelvin probe force microscopy[37,38]. Moreover, for MoS$_2$ grown on Au(111), S vacancies

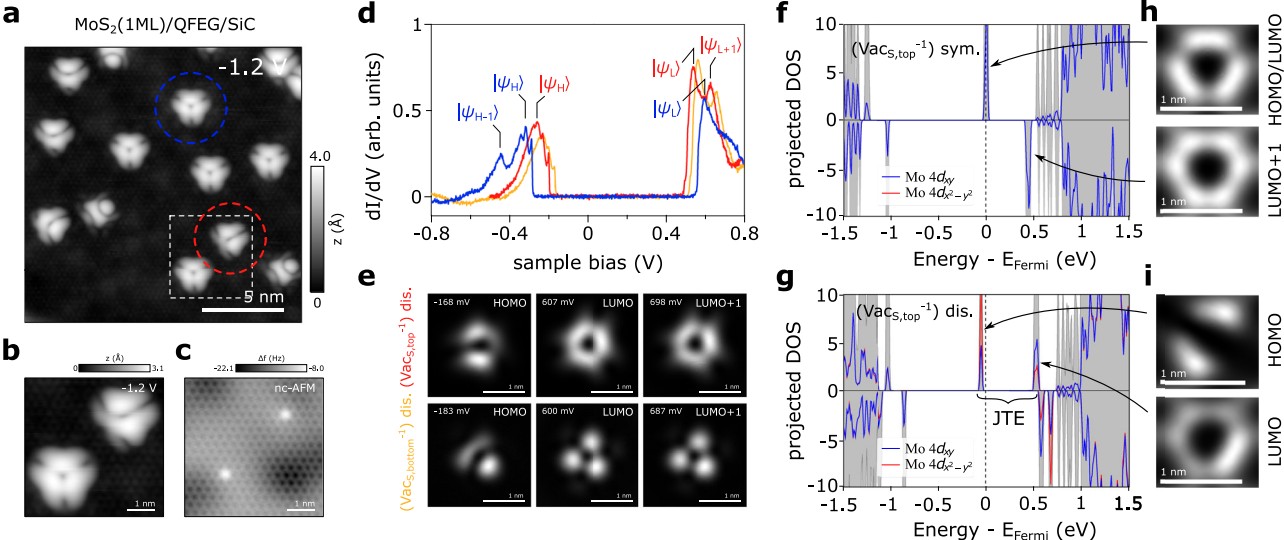

**Fig. 1 | Symmetric and symmetry-broken (distorted) negatively charged sulfur vacancies (Vac$_S^{-1}$). a** Scanning tunneling microscopy (STM) topography ($I = 100$ pA) of annealing-induced Vac$_S^{-1}$ (colored circles) in monolayer MoS$_2$ on quasi-freestanding epitaxial graphene (QFEG) on SiC. **b** Zoom-in STM topography ($I = 100$ pA) of the white dashed box in (**a**) containing a distorted (top right) and a symmetric (bottom left) Vac$_S^{-1}$. **c** Noncontact atomic force microscopy (nc-AFM) image with a metallic tip of the same area as in (**b**) proves that both defects are Vac$_S$ in the upper S layer. **d** d$I$/d$V$ spectra of the frontier in-gap defect orbitals (HOMO/HOMO-1: $|\Psi_{H,H-1}\rangle$ and LUMO/LUMO+1 $|\Psi_{L,L+1}\rangle$) of the symmetric (blue) and distorted (red)

Vac$_{S,top}^{-1}$, as well as the distorted Vac$_{S,bottom}^{-1}$ (orange) (lock-in amplitude: $V_{mod} = 2$ mV). **e** Corresponding constant height d$I$/d$V$ maps of the distorted S top (upper row) and bottom (bottom row) vacancies highest occupied (HOMO) and lowest unoccupied defect orbitals (LUMO, LUMO+1), labeled in (**d**) ($V_{mod} = 20$ mV). Projected density of states (pDOS) of Vac$_S^{-1}$ in MoS$_2$ in the symmetrized (**f**) and distorted (**g**, ground state) geometry, calculated in a $4 \times 4$ supercell. The pDOS onto the Mo $d_{xy}$ and $d_{x^2-y^2}$ orbitals are shown in red. The energy splitting due to the Jahn-Teller effect (JTE) is indicated. Constant height DOS map of the HOMO and LUMO(+1) orbitals for the symmetrized (**h**) and distorted (**i**) Vac$_{S,top}^{-1}$ geometry.

have recently been shown to exhibit a Kondo resonance[38], originating from an unpaired electron spin associated with the negative charge of the vacancy.

While some $Vac_S^{-1}$ appear threefold symmetric, others unexpectedly exhibit a 'distorted' appearance with only a two-fold symmetry at negative sample bias (Fig. 1a, b), even though both type of defects appear in nc-AFM as sulfur vacancies in the top sulfur layer (Fig. 1c). Also in scanning tunneling spectroscopy (STS) measurements the electronic structure between the symmetric and distorted vacancy species are distinct as shown in Fig. 1d. The symmetric $Vac_S^{-1}$ has two occupied in-gap defect states (HOMO-1 and HOMO) and one LUMO resonance close to the Fermi energy (blue), while the distorted $Vac_S^{-1}$ features one HOMO resonance (with vibronic side-bands) and two LUMO resonances (red and orange). The symmetric $Vac_S^{-1}$ preserves the underlying $C_{3v}$ lattice symmetry in its frontier orbitals (see Supplementary Fig. 4e, f), whereas the HOMO of the distorted $Vac_S^{-1}$ variant has a lower $C_{2v}$ symmetry (see Fig. 1e). Such a behavior was not observed for $Vac_S^0$ in $WS_2$[33], which indicates the decisive role of the defect charge state.

We attribute the reduced point group symmetry of $Vac_S^{-1}$ to a JTE as previously predicted[22]. Upon charging, the degenerate S vacancy state (Fig. 1f) becomes unstable and relaxes, thereby lifting the orbital degeneracy (Fig. 1g). The distorted $Vac_S^{-1}$ geometry (Supplementary Fig. 12c–e) has a lower total energy by 142 meV[22]. The calculated density of states map of the frontier orbitals shown in Fig. 1i are in excellent agreement with the experimental differential conductance (d$I$/d$V$) maps in Fig. 1e. However, the reason for the co-existence of both the symmetric and distorted $Vac_S^{-1}$ is not entirely clear. Possibly, the symmetric variant might be a doubly negatively charged vacancy $Vac_S^{-2}$, which is not Jahn-Teller active[22]. Importantly, these findings are not limited to $MoS_2$, but similarly apply to the negatively charged S vacancy in $WS_2$ and negatively charged Se vacancy in $WSe_2$ (see Supplementary Fig. 15), despite their significantly stronger SOC.

Motivated by the surprising discovery of symmetry breaking in one of the best studied defects in TMDs, we designed a defect system that we can stabilize in different charge states by modulating the substrate chemical potential. Next we will discuss the charge state tristability of rhenium dopants in $MoS_2$ and their charge state-dependent symmetry breaking. Rhenium substituted for molybdenum ($Re_{Mo}$) has a $D_{3h}$ point symmetry group and produces fundamentally different in-gap states as compared to $Vac_S$, which makes it an ideal candidate to test the generality of our findings.

## Charge state tristability of Re Dopants in $MoS_2$

Chemical doping of 2D semiconductors has been a popular but challenging route to control their conductivity due to the large defect ionization energies associated with the tightly confined defect wave functions in 2D[39]. Recently, a scalable MOCVD process has been developed to introduce Re dopants in TMDs in controllable concentrations from hundreds of ppm to percentage concentrations[40,41]. Substitutional Re dopants in $WSe_2$ on QFEG are positively charged (ionized) and feature a series of closely-spaced unoccupied defect states below the conduction band minimum. Here we leverage the higher electron affinity $\chi$ of monolayer $MoS_2$ as compared to $WSe_2$, to position the Fermi level in-between these $Re_{Mo}$ defect states. In $MoS_2$/QFEG, we find that most Re impurities are charge neutral while some remain positively charged, which also applies to Re-doped bilayer $MoS_2$ grown on QFEG. By using epitaxial graphene (EG) without hydrogen intercalation as a substrate, which has a substantially smaller work function than QFEG, we push the Fermi level even higher[42,43]. As a consequence, this results in the Re impurities becoming negatively charged, i.e., they accept an electron, somewhat counter-intuitive for an $n$-type dopant. By varying the substrate chemical potential, we establish a charge state tristability of Re dopants in $MoS_2$.

In Fig. 2b we present a large-scale STM topography of Re-doped (5%) $MoS_2$ on partially H-intercalated EG. In areas with H-intercalation (gray plateau in Fig. 2b and structural model shown in Fig. 2c), we observe the Re dopants in the positive ($Re_{Mo}^{+1}$) or neutral ($Re_{Mo}^0$) charge state, whereas on EG substrates (blue canyon in Fig. 2b, and structural model in Fig. 2c) all Re are negatively charged ($Re_{Mo}^{-1}$). The charge states can be discriminated by the characteristic upwards (anionic impurity) or downwards (cationic impurity) band bending in STS. This translates into a dark depression (Fig. 2a left) or bright protrusion (Fig. 2a right) of the STM topography if measured at the conduction band edge. Most importantly, $Re_{Mo}^{+1}$ preserves the underlying $D_{3h}$ lattice symmetry, as previously reported for $Re_W^{+1}$ in $WSe_2$[40], while the STM topography of $Re_{Mo}^0$ and $Re_{Mo}^{-1}$ appear slightly distorted. This is consistent with their imaged defect orbital symmetries discussed next.

The pink d$I$/d$V$ spectrum in Fig. 2e and defect orbitals depicted in Supplementary Fig. 5 for $Re_{Mo}^{+1}$ closely resemble the $Re_W^{+1}$ states observed in $WSe_2$[40], with several defect states just above the Fermi level. $Re_{Mo}^0$ (Fig. 2e blue, and Supplementary Fig. 6) exhibits several unoccupied defect states above Fermi, along with an unusually broad defect resonance below it. Lastly, $Re_{Mo}^{-1}$ shown in green in Fig. 2e and Supplementary Fig. 7, has only a broad defect resonance below the Fermi level. Based on the STS spectra we derive a simplified schematic level diagram for the differently charged $Re_{Mo}$ impurities in Fig. 2d. Note that for $Re_{Mo}^0$ we only draw one level at the Fermi energy despite two STS resonances being observed around the Fermi level. Due to the Coulomb energy associated with adding or removing an electron to or from a singly occupied orbital, we observe two resonances both in the occupied and unoccupied state[44,45]. The EG and QFEG act as a Fermi sea where electrons can be donated to or withdrawn from the $Re_{Mo}$. Upon (de)charging, many-body effects renormalize the energies of all defect states because of their pronounced localization. For instance, the unoccupied defect states for $Re_{Mo}^{-1}$ are pushed higher in energy, whereas they become closely spaced at low energy for $Re_{Mo}^{+1}$.

## Symmetry breaking of $Re_{Mo}^{-1}$ and $Re_{Mo}^0$

To probe whether the striking electronic asymmetry of $Re_{Mo}^0$ and $Re_{Mo}^{-1}$ results from a (measurable) lattice distortion, we use CO-tip nc-AFM[27] as shown in Fig. 3d–f. Due to the high surface sensitivity of CO-tip nc-AFM, only the surface S atoms can be resolved as bright (repulsive) features. Mo atom positions can be inferred from the slightly more attractive interaction as compared to the hollow site, establishing the full unit cell of 1H-$MoS_2$ indicated as an inset in Fig. 3d–f. Although the Re impurity itself cannot be directly resolved, its location (marked by the colored triangle) can be deduced from correlation with the STM topography (Fig. 3a–c). For $Re_{Mo}^0$ and $Re_{Mo}^{-1}$ we find that one of the neighboring S atoms appears brighter, indicating an outwards relaxation from the S plane, which can be seen in the line-cut below the respective nc-AFM image. Occasionally, the protruding S atom is found to switch during nc-AFM measurements at close tip–sample distance (Supplementary Fig. 8). For $Re_{Mo}^{+1}$ no such relaxation could be resolved. Apart from the vertical relaxation, which is most pronounced for $Re_{Mo}^{-1}$, also a lateral relaxation of the two opposing S atoms is observed (Fig. 3g). The vertical protrusion and lateral strain profile are in excellent agreement with the simulated CO-tip nc-AFM image (Fig. 3h) based on the calculated relaxed geometry shown in Fig. 3i[46].

We employ DFT with a 2D charge correction scheme[47] to investigate the atomic and electronic structure of $Re_{Mo}$ in $MoS_2$, considering the positive, neutral, and negative charge states. While earlier ab initio studies failed to capture the symmetry breaking in the neutral defect state[48–50], our analysis demonstrates that enforcing integer electronic occupation successfully reproduces a distorted ground state geometry for $Re_{Mo}^0$ and $Re_{Mo}^{-1}$. In Fig. 4a–c, we present the adiabatic potential energy surface (APES) for all three charge states. In the positive charge state, the APES exhibits a single minimum at the high

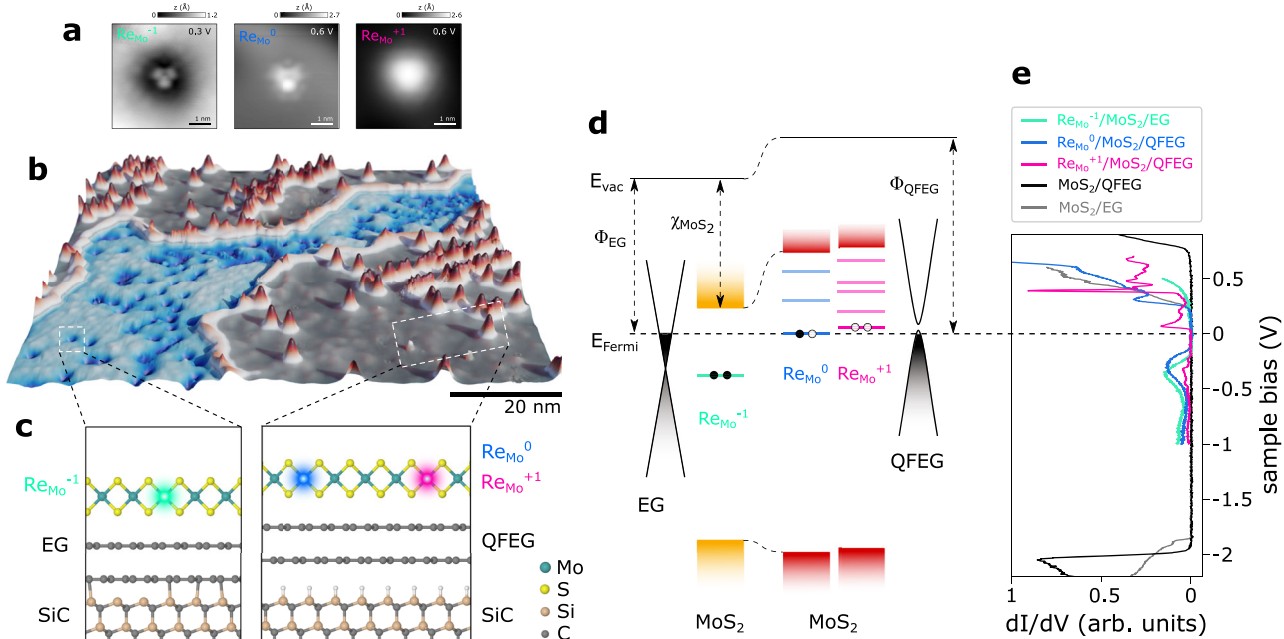

**Fig. 2 | Charge state tristability of Re dopants in MoS₂ on EG and QFEG on SiC.**
**a** STM topographies ($I = 100$ pA) of single Re dopants in the three different charge states ($Re_{Mo}^{-1}$, $Re_{Mo}^{0}$, and $Re_{Mo}^{+1}$). **b** 3D representation of a large-scale STM image of Re-doped monolayer MoS₂ on domains of epitaxial graphene (EG) (center blue region) or domains of QFEG substrate (outer gray region). At positive bias (here 0.6 V), Re dopants are imaged as a dark blue pits on EG whereas they appear as red hills on MoS₂ grown on QFEG, depending on their charge states. **c** Atomic model of the MoS₂/EG and MoS₂/QFEG interface. On EG only $Re_{Mo}^{-1}$ are observed, whereas on

QFEG both $Re_{Mo}^{0}$ and $Re_{Mo}^{+1}$ are found. **d** Schematic electronic level diagram of the MoS₂/EG (left) and MoS₂/QFEG interface (right) with the $Re_{Mo}$ level occupation indicated. The work function of EG $\Phi_{EG}$ is substantially smaller than the work function of QFEG $\Phi_{QFEG}$. $\chi_{MoS_2}$ denotes the electron affinity of MoS₂. $E_{vac}$ and $E_{Fermi}$ denote the vacuum and Fermi level, respectively. **e** d$I$/d$V$ spectra of $Re_{Mo}^{-1}$ (green), $Re_{Mo}^{0}$ (blue), and $Re_{Mo}^{+1}$ (pink), pristine monolayer MoS₂ on EG (gray) and on QFEG (black), respectively. Bias modulation amplitude: $V_{mod} = 20$ mV.

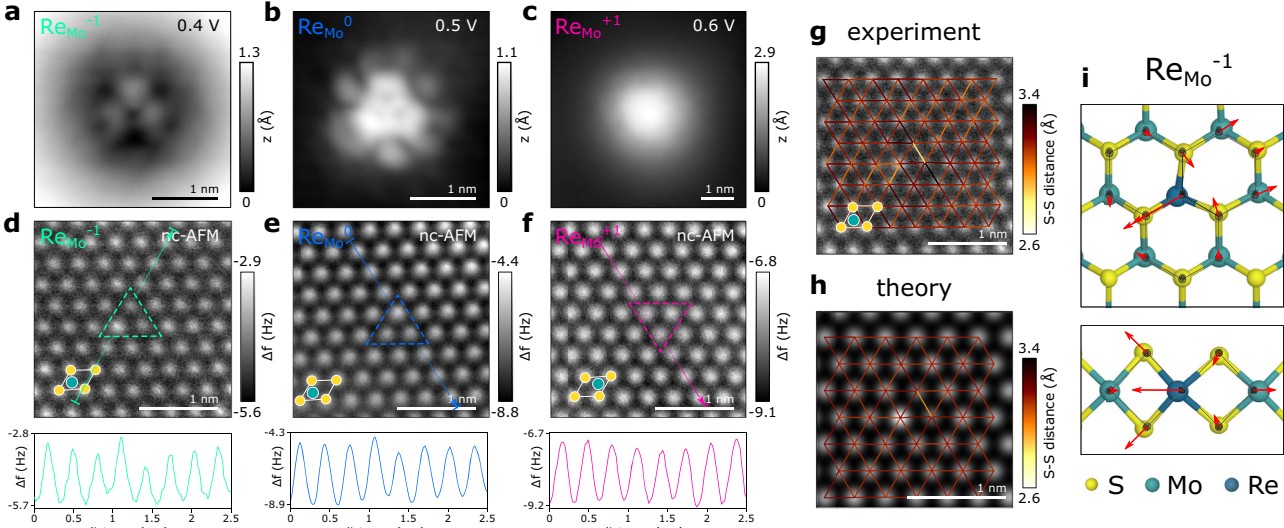

**Fig. 3 | Charge state-dependent Jahn-Teller distortions. a–c** STM topography ($I = 100$ pA) of $Re_{Mo}^{-1}$, $Re_{Mo}^{0}$, and $Re_{Mo}^{+1}$, respectively. **d–f** CO-tip nc-AFM image of the same Re impurity as shown in (**a–c**). Top sulfur atoms are repulsive (bright) and molybdenum atoms attractive (dark). The MoS₂ unit cell is displayed at the bottom left in each image (S: yellow, Mo: turquoise). The Re atoms are located in the center of the dashed triangle, bound to the three bright S atoms on the top surface. A Δ$f$ line profile across a sulfur row (arrow in image) is shown in the lower panel, highlighting a protruding S atom next to the Re impurity for $Re_{Mo}^{-1}$ and slightly weaker

for $Re_{Mo}^{0}$. Measured (**g**) and simulated (**h**) CO-tip nc-AFM image of $Re_{Mo}^{-1}$. The apparent sulfur-sulfur distances in the top S layer is indicated by the red-to-yellow lines. Measurement and simulation show significant lateral compressive strain introduced by the Re impurity at nearest-neighbor S atoms opposite to the protruding S atom. **i** Calculated geometry of $Re_{Mo}^{-1}$ exhibiting considerable distortions. The red arrows indicate the atom relaxations as compared to the pristine MoS₂ lattice (gray ball and stick model). For clarity, the arrows are ten times longer than the actual relaxation.

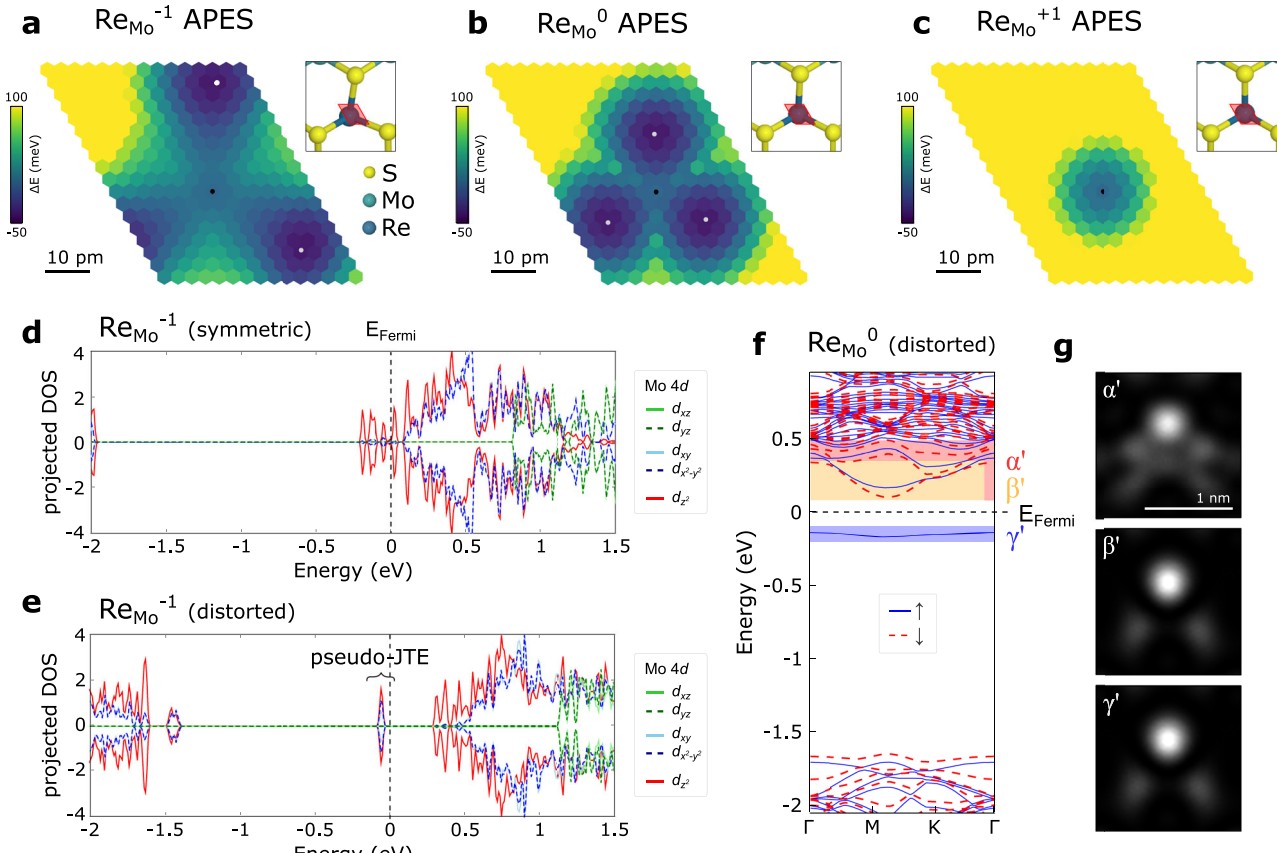

**Fig. 4 | Adiabatic potential energy surface (APES) and projected DOS (pDOS) of Re$_{Mo}$ in different charge states. a–c** Potential energy surface of the Re dopant for the negative, neutral, and positive charge state respectively. The energies correspond to the Γ point energy in a 5 × 5 supercell. The black and white dots indicate the symmetric and distorted positions of Re, respectively. The sampled area for the APES is marked by a red parallelogram in each structural model inset. **d, e** Projected density of states of Re$_{Mo}^{-1}$ in the constrained symmetric (**d**) and distorted ground state geometry (**e**), exhibiting electronic reconfiguration resulting from the pseudo-JTE. **f** Spin-polarized Re$_{Mo}^{0}$ band structure resulting in a paramagnetic ground state. **g** Constant height DOS maps of different energy windows in the Re$_{Mo}^{0}$ band structure, as indicated in (**f**). The $\alpha'$ image reflects the STM contrast at low bias. The $\beta'$ and $\gamma'$ images correspond to the lowest unoccupied and highest occupied Re$_{Mo}^{0}$ orbital, respectively (cf. Fig. 5b C3 and D1).

symmetry position. However, for Re$_{Mo}^{0}$ and Re$_{Mo}^{-1}$, three equivalent off-centered minima are observed at distances of 13 pm and 25 pm, respectively, towards one of the neighboring S atoms. Although the structural relaxation of Re away from the center is twice as large for the negative charge state, the energy gain of approximately −60 meV is comparable for both neutral and negative charge states. The obtained ab initio APES supports the experimentally observed symmetry-broken ground states of Re$_{Mo}^{0}$ and Re$_{Mo}^{-1}$.

To gain insight into the driving force behind the observed symmetry breaking, we examine the orbital-projected density of states (pDOS) of Re$_{Mo}^{-1}$ in both the symmetric and distorted ground state geometries, as shown in Fig. 4d and e, respectively. In contrast to the Vac$_S^{-1}$ case, the symmetric configuration of Re$_{Mo}^{-1}$ does not occupy a degenerate orbital at the Fermi energy. Instead, two electrons occupy the non-degenerate $d_{z^2}$ HOMO (red curve in Fig. 4d). The excited state spectrum reveals degeneracies with dominant contributions from $d_{xz}$ and $d_{yz}$ (green solid and dashed line) orbitals and states composed of $d_{xy}$ and $d_{x^2-y^2}$ (blue solid and dashed line) orbitals. The structural distortion lowers the energy of the HOMO by creating a hybridized $d_{z^2} - d_{x^2-y^2}$ orbital that is doubly occupied and shifts down to the mid-gap region. A priori it is unclear if the hybridized orbital results from occupying a $d_{z^2}$ state that takes on $d_{x^2-y^2}$ orbital character, or vice versa, which would either suggest that the mechanism is attributed to a pseudo-JTE or hidden-JTE, respectively. To differentiate between a pseudo-JTE and a hidden-JTE, one can look at the energy along the distortion direction in the APES (Fig. 4a). A hidden-JTE would typically

manifest as an energy barrier between the symmetric and distorted geometry, with the initially occupied state experiencing an energy increase that is eventually surpassed by the energy gain of the previously unoccupied state[19]. Consequently, the absence of such an energy barrier confidently identifies the distortion mechanism in this system as being governed by the pseudo-JTE.

Similarly, the distortion in the neutral charge state is also caused by the pseudo-JTE (cf. Supplementary Fig. 17). In the neutral charge state, the odd number of electrons gives rise to a spin-polarized ground state, wherein a singly occupied $d_{z^2} - d_{x^2-y^2}$ SOMO is established (Fig. 4f). This leads to a net magnetic moment that is absent in the positive or negative charged defects. The presence of a paramagnetic ground state in the neutral defect with a singly occupied spin channel is not captured by previous ab initio descriptions of Re$_{Mo}^{0}$ in MoS$_2$[49]. However, experimental studies using electron paramagnetic resonance have successfully detected Re doping in natural MoS$_2$ samples, providing further validation for our findings[51]. Furthermore, the spatial distribution of the highest occupied ($\gamma'$) and lowest unoccupied ($\beta'$) Re$_{Mo}^{0}$ orbitals shown in Fig. 4g exhibit the same spatial distribution in perfect agreement with the experimental d$I$/d$V$ images, as shown in Fig. 5b (resonances C3 and D1). Therefore, we conclude that the symmetry-broken paramagnetic ground state observed in the neutral Re$_{Mo}$ substitution reconciles our theoretical predictions with experimental observations based on scanning probe microscopy and previous measurements using electron paramagnetic resonance.

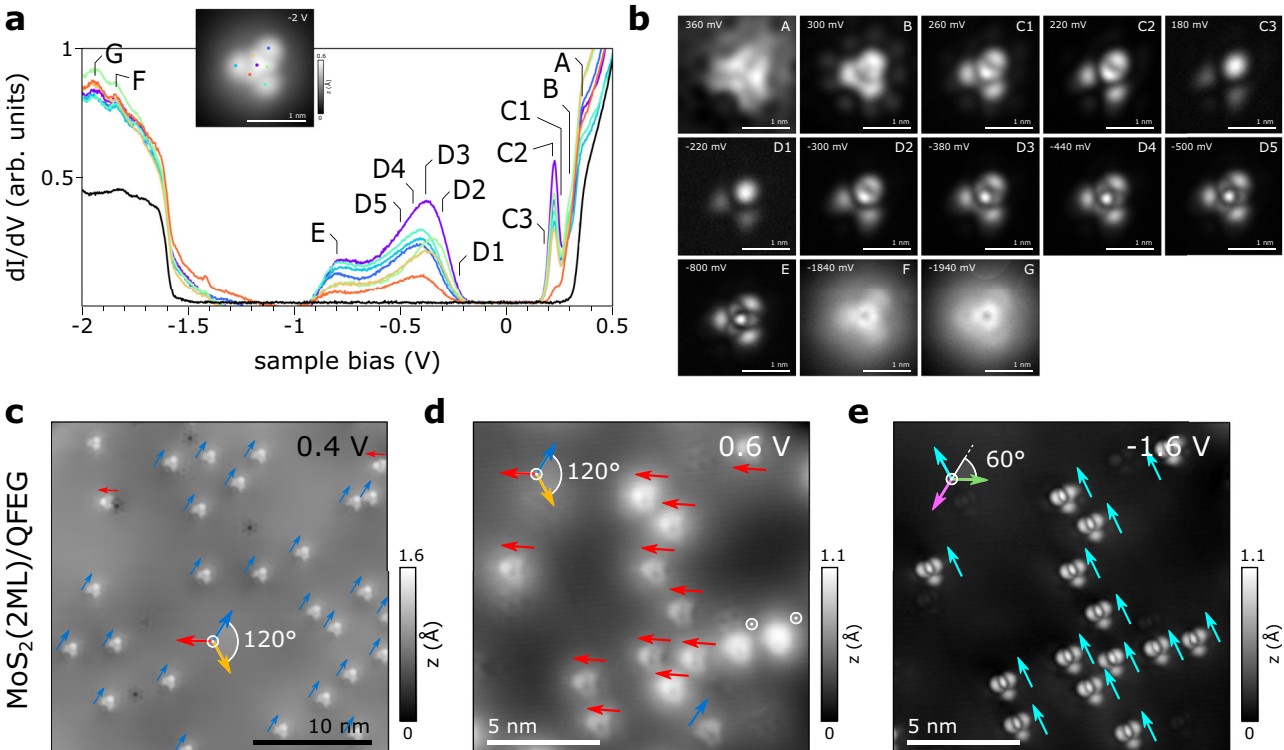

**Fig. 5 | Configurational continuum and directionally aligned domains. a** d$I$/d$V$ spectroscopy of $Re_{Mo}^0$ on bilayer $MoS_2$ ($V_{mod}$ = 10 mV). The spectroscopy locations on the defect are indicated with a dot of the same color in the inset STM image, and the pristine $MoS_2$ (2 ML) spectrum is shown in black. **b** Constant height d$I$/d$V$ maps ($V_{mod}$ = 20 mV) at the sample voltages labeled in (**a**). A progression of the orbital image is observed within certain resonances (cf. D1-D5) that might explain their large broadening. The resemblance of the highest occupied (D1) and lowest unoccupied defect orbital (C3) is indicative of a singly occupied defect state. **c** STM topography ($I$ = 50 pA) of neutral Re dopants (top layer) in bilayer $MoS_2$ on QFEG. The orientation of the distorted STM contrast along the crystal axes is indicated by colored arrows. **d, e** STM topography ($I$ = 100 pA) of $Re_{Mo}^0$ in bilayer $MoS_2$ in a different location also showing a preferential orientation. Here, the $D_{3h}$ symmetry breaking is observed at positive and negative bias, however, the respective symmetry axis is canted by 60°.

## Configurational continuum and directional alignment

The unusually broad defect resonances on the order of 300 meV observed for some $Re_{Mo}^0$ and $Re_{Mo}^{-1}$ states (Fig. 2e blue and green) raises the question about the broadening mechanism. For many TMD defects, the intrinsic lifetime broadening primarily caused by the weakly interacting graphene substrate, is only a few meV[33]. We do not expect a qualitatively different substrate interaction for Re dopants, as it is mainly influenced by the distance to the substrate, and the measured defect state broadening does not decrease for Re in bilayer $MoS_2$ (Fig. 5a). In low-temperature tunneling spectroscopy of single defects in 2D semiconductors, significant broadening of electronic resonances can occur due to strong electron-phonon coupling[45]. However, in most cases the confined electronic states couple strongly only to a small set of localized modes with energies of a few tens of meV, given the weight of the atoms involved[45]. Therefore, it is typically possible to differentiate the vibronic ground state resonance (zero-phonon line) and the first few vibronic excited states with an intrinsic broadening of only 3−4 meV, as can be seen for instance in the d$I$/d$V$ spectra of the sulfur vacancy in Fig. 1d. Hence, vibronic broadening is unlikely the main reason for the large defect state broadening.

Remarkably, we observe that the orbital image of the broad defect state resonance undergoes a continuous evolution of contrast, depending on the energy at which the defect state is probed, as shown in Fig. 5a, b. This behavior is observed on both mono- and bilayer $MoS_2$, although the direction of the progression varies depending on the precise location within the sample (cf. series Fig. 5b and Supplementary Fig. 6c). This unusual behavior might be a consequence of the comparably flat APES introduced by the pseudo-JTE[18,52], establishing a configurational continuum without significant energy barrier between

the states (Fig. 4a–c). This phenomenon is common for Jahn-Teller systems and is typically referred to as dynamic JTE[18,52]. Such a configurational continuum may explain the unusual progression of the STS contrast within the defect resonance and its significant broadening.

In most regions of the sample, the symmetry breaking direction is not a random distribution. Rather, there are larger domains where all $Re_{Mo}^0$ or $Re_{Mo}^{-1}$ relax in the same direction, as illustrated in Fig. 5c–e. Directionally-aligned domains have been observed not only in highly doped (5%) samples, but also in samples with comparably low doping concentrations (<0.1%). Therefore, we suspect that residual strain in the epitaxially grown TMD, rather than direct dopant-dopant interactions, could be responsible for the aligned domains. The flat APES may render the symmetry-broken Re particularly susceptible to external perturbations such as strain. In turn, this susceptibility may serve as a local strain sensor. Further studies are needed to establish a firm correlation between the mesoscale strain tensor and local Jahn-Teller distortion of the individual Re dopants.

In summary, we have directly visualized the charge state-dependent symmetry breaking at S vacancies and Re substituting for Mo in mono- and bilayer $MoS_2$. $Vac_S^{-1}$ (in the top and bottom) coexists in both a threefold symmetric and symmetry-broken state. By controlling the Fermi level alignment through the substrate chemical potential, we are able to stabilize Re in three different charge states. While $Re_{Mo}$ in the positive charge state retains the underlying $D_{3h}$ lattice symmetry, $Re_{Mo}^0$ and $Re_{Mo}^{-1}$ exhibit an increasingly distorted geometry. We assign the minute local lattice relaxation detected by CO-tip nc-AFM and significant electronic reconfiguration of the defect states measured by STS to a pseudo-JTE, corroborated by DFT calculations. A progressive defect orbital evolution observed

within the unusually broad defect resonances is attributed to a flat potential energy surface, which leads to a continuum of the ground state geometry configuration. Complementary measurements are needed to corroborate the link between the intriguing observation of directionally aligned Re domains and an inherent strain profile in the $MoS_2$ layers. The charge-state dependent JTE exhibited by chalcogen vacancies and Re dopants offers a powerful means to precisely control the photo-physics of these prototypical atomic quantum emitters.

## Methods

### Sample preparation

Metal-organic chemcial vapor deposition was used to grow monolayer $MoS_2$ on an epitaxial graphene and quasi-freestanding epitaxial graphene on 6H-SiC(0001) substrate. $H_2S$ and $Mo(CO)_6$ were used as precursors for growth of $MoS_2$, while the Re dopants were introduced by admixing 0.1%, 5% of $Re_2(CO)_{10}$ during the growth of $MoS_2$. Before SPM inspection, all samples were annealed to 300 °C for about 30 min to remove the possible adsorbates. S vacancies in $MoS_2$ were created by annealing the pristine $MoS_2$ to 900 °C for 30 min. Epitaxial graphene was synthesized *via* silicon sublimation from the Si face of SiC[53], subsequently monolayer epitaxial graphene is annealed at 950 °C for 30 min in pure hydrogen to intercalate hydrogen between the buffer layer and silicon carbide, yielding bilayer quasi-freestanding epitaxial graphene[54].

### Scanning probe microscopy (SPM) measurements

SPM measurements were acquired with a Scienta-Omicron GmbH or CreaTec Fischer & Co. GmbH scanning probe microscope at liquid helium temperatures ($T < 5$ K) under ultrahigh vacuum ($p < 2 \times 10^{-10}$ mbar). The quartz crystal cantilever (qPlus based) sensor[55] ($f_0 \approx 27$ kHz, $Q \approx 47 k$) tip apex was prepared by indentations into a gold substrate and verified as metallic on a Au(111) surface. Non-contact AFM images were taken with a carbon monoxide functionalized tip[56] in constant height mode at zero bias with an oscillation amplitude $A_{osc} < 100$ pm. STM topographic measurements were taken in constant current feedback with the bias voltage applied to the sample. STS measurements were recorded using a lock-in amplifier (HF2LI Lock-in Amplifier from Zurich Instruments or Nanonis Specs) with a resonance frequency between 600 and 700 Hz and a modulation amplitude provided in the figure caption.

### First-principles calculations

All calculations were performed using DFT as implemented in the Vienna ab initio simulation package VASP[57]. We modeled the core electrons using projector-augmented wave (PAW) potentials[58,59] with valence electron configurations of $4p^64d^55s^1$ and $3s^23p^4$ for Mo and S, respectively. The cutoff energy was 520 eV, which ensures energy convergence to within 1 meV/atom, and all calculations were set to be spin-polarized unless SOC was considered. The defect systems were all modeled as $5 \times 5 \times 1$ supercells with 20 Å vacuum spacing between layers with a single defect per supercell. In addition, we enforced integer band occupancies during our calculations to avoid unphysical electronic ground states with fractional band occupancies, which may arise due to wavefunction overlap. The only case where integer band occupancies were not enforced was during the symmetry-constrained $Vac_S^{-1}$ calculation. Fractional occupancies were allowed to highlight the symmetry breaking between the $d_{xy}$ and $d_{x^2-y^2}$ states.

In all calculations aside from the APES, we used a $5 \times 5 \times 1$ Γ-centered Monkhorst-Pack k-point mesh[60]. We also treated the exchange-correlation in calculations aside from the potential energy surface using the strongly constrained and appropriately normed meta-generalized gradient approximation functional with van der Waals interactions (SCAN+rVV10)[61]. In the calculations used to create

the APES, the computational cost was reduced by using a Γ point-only calculation in the PBE functional.

The electronic structure calculations were performed using a $5 \times 5 \times 1$ Γ-centered Monkhorst-Pack k-point mesh and a Gaussian smearing with a smearing parameter of 0.01 eV.

## Data availability

Relevant data supporting the key findings of this study are available within the article and the Supplementary Information file. All raw data generated during the current study are available from the corresponding authors upon request.

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

## Acknowledgements

We thank Adam Gali for valuable discussions. F.X. acknowledges the Walter Benjamin Program from Deutsche Forschungsgemeinschaft (DFG) and the Swiss National Science Foundation (Grant No. 210093). L.H., J.A and B.S. appreciate funding from the European Research Council (ERC) under the European Union's Horizon 2020 research and innovation program (Grant agreement No. 948243). R.T., Y.-C.L. and J.A.R. acknowledge funding from NEWLIMITS, a center in nCORE as part of the Semiconductor Research Corporation (SRC) program sponsored by NIST through award number 70NANB17H041 and the Department of Energy (DOE) through award number DESC0010697. R.T., C.D. and J.A.R. acknowledge funding from the 2D Crystal Consortium, National Science Foundation Materials Innovation Platform, under cooperative agreement DMR-1539916. For the purpose of Open Access, the author has applied a CC BY public copyright license to any Author Accepted Manuscript version arising from this submission.

## Author contributions

B.S. and J.A.R. conceived the idea. R.T., C.D., Y.-C.L. and J.A.R. grew the samples. F.X., L.H., J.A., and B.S. carried out the experiments. P.A.V., A.M.T. and R.G.H. performed the DFT calculations. P.R., R.F. and O.G. helped supervise the project. F.X., P.A.V. and B.S. wrote the manuscript. All authors discussed the results and contributed to the final manuscript.

## Competing interests

The authors declare no competing interests.
