## [Peer Review File · Nature Communications]

Charge State-Dependent Symmetry Breaking of Atomic Defects in Transition Metal DichalcogenidesEditorial Note: Parts of this Peer Review File have been redacted as indicated to remove third-party material where no permission to publish could be obtained.

REVIEWER COMMENTS

Reviewer #1 (Remarks to the Author):

Xiang and co-authors report the observation of charged defects on a transition metal dichalcogenide. They claim that they image a broken symmetry state by means of Scanning Tunneling Microscopy and Spectroscopy. While I believe that the experimental data are a beautiful characterization of the TMD defects and the experiments are carefully performed, I do not believe the current manuscript significantly contributes to the advancement of the field. I also believe that the interpretation of the data in terms of broken symmetries is wrong. Below I list my concerns.

- The main claim is the observation of symmetry breaking. However, what they observe is a defect with a particular orbital distribution. First, discrete symmetries are described in crystals. A defect is not the crystal, it does not follow the crystalline lattice – that’s why it is a defect. One could argue that the defect breaks translational symmetry, but that does not mean much in terms of the underlying physics for the material itself (unless its properties are determined by disorder). Moreover, it wouldn’t be a spontaneous symmetry breaking if it happens through lattice distortions (whether is a defect or strain).
- As far as I understood, the observed two-fold LDOS on the defects comes from the orbital splitting. As a consequence, some orbitals are filled, some others are empty, and their distribution is two-fold instead of three-fold. That does not break the symmetry, it actually follows the symmetry of the orbitals (or a combination of them for that matter).
- In general, I think the authors fail on clearly explaining what new physics we learn from their experimental findings, what the significance is.
- On a side note, the authors claim to do chemical gating and to ‘control the Fermi level alignment through the substrate chemical potential’. How do the authors control that? It is phrased in a way that it seems like they gate the system, but I think that it’s just the natural doping from the fact that the MoS₂ is on top of a substrate. If they don’t control the doping with an external voltage through an insulating layer, they should not use the term ‘gating’.

Reviewer #2 (Remarks to the Author):

In their article “Charge State-Dependent Symmetry Breaking of Atomic Defects in Transition Metal Dichalcogenides”, Feifei Xiang and coworkers combine state-of-the-art scanning probe experiments with DFT calculations to investigate 2 types of defects in CVD-grown monolayer MoS₂. For the quite common S vacancies the authors find two different configurations once the defect level is charged forming VacS-1 confirming a prediction which was made in Ref. 22 of the manuscript. The authors then continue to

investigate Re dopants as those can be prepared in different charged states in contrast to the VacS. And indeed, also for the ReMo-1 the authors find a broken-symmetry ground state. While the symmetry-broken S vacancy is due to a “normal” Jahn-Teller distortion, the authors find that the symmetry reduction for the Re dopants is due to a pseudo Jahn-Teller effect.

I find the manuscript well written and the results convincing. Furthermore, I think that these results are really interesting for a lot of people working on 2D materials and it could be published. Yet, the distortions found in MoS₂ might be also specific to this system and thus, in order to justify better the publication in Nat. Commun., I think the authors should add a paragraph at the end, if the distortions can also be expected in other systems such as the other semiconducting 2H-TMDCs or -even better - check for, e.g., WSe₂.

Otherwise I have only a few minor points concerning the calculations which however also need to be addressed before any publication.

- at the end of page 5, the authors write (when speaking about the possibility of having VacS-2:

“Possibly, the system is trapped in a local minimum with a potential barrier too high to relax to the global potential energy minimum.”

This could be checked maybe via CI-NEB, or at least a mapping as shown in Fig. 4 for the Re dopant. Especially checking along a linear interpolation of the geometries of the symmetric/distorted structures.

- Why did the authors not include SOC in their calculations? It is known to be important in TMDCs and also the defect states in WS₂ and WSe₂ split in two separate bands due to SOC as shown in, e.g., Ref. 34 and <https://doi.org/10.1038/s41699-023-00421-0>, respectively. Even if the Mo-based TMDCs have much lower SOC splitting, I think the authors should also check if the inclusion of SOC changes some of their conclusion.

- A 4x4 supercell seems to be rather small, especially for a charged defect. In similar calculations for uncharged chalcogen vacancies, I found that there was still a considerable interaction between the repeated defects. The authors should check that their results do not depend on an increase to 5x5 or need to discuss the possible interaction at least.

- The k-point grids are only given for the DOS calculations. The grids for the relaxation is needed as well. Why 6x6x2 for a 2D system? (why 2 in z direction) Also, why did the authors use the tetrahedron method in one case but a Gaussian smearing in the other? Also, why the different functionals for the different defects? I find it surprising that for the 2 defects so many parameters were changed – also the k-point density (for the DOS calculations) are different.

Reviewer #3 (Remarks to the Author):

Review of "Charge State-Dependent Symmetry Breaking of Atomic Defects in Transition Metal Dichalcogenides" by Feifei Xiang et. al

Noteworthy results:

- Impact of substrate choice on the Fermi level placement of MoS₂ and charged states of Re dopants.
- Systematic STS study on sulfur vacancies and Re dopants on MoS₂ providing insight on Re multiple stable charge states and sulfur vacancies symmetry.

Relevance:

- This work is significant to the field, as it thoroughly characterizes doped MoS₂ and discusses the impact of different charge dopants and their symmetry on MoS₂ electronic properties. To the best of my knowledge and search, there are no systematic dI/dV mapping studies on doped MoS₂ that discuss charge state and symmetry breaking. This work corroborates to the advance of knowledge that impact tunability of optical emission properties and spin-photon interfaces for future electronic applications.

The manuscript is well-written, and the topic relevant. However, there are some points to be discussed and clarified before I can make the recommendation for this manuscript to be published fitting the high standards of Nature Communications journal:

- 1) How were the intrinsic defects identified? The authors claim that O_S, CH_S, Cr_{Mo} and W_{Mo} are also present in the sample, apart from the studies Vac_S, but no convincing method was described as to how each type of intrinsic defect was properly identified. Are there simulated STM images that support these claims? Otherwise, the somehow arbitrary identification of defects is not convincing.
- 2) On the Re doping study, again a better argument is needed to guarantee that those were Re substitutionals. How can the authors be confident that they are not defects from other nature, such as the ones previously cited. Simulated STM images would again help with this discussion point.
- 3) The authors mentioned in page 6 that "Counter-intuitive for an n-type dopant, this results in the Re impurities becoming negatively charged, i.e. they accept an additional electron.". Since it is counter-intuitive, could the authors present a better explanation for why the charge transfer is still happening. Forgive me if the explained that somewhere else if I missed it. It would still be beneficial to include this discussion directly following the statement.
- 4) Since the STM topography image on fig. 3(c) presents a very strong contrast difference and lacks atomic resolution, it is an overstatement to rely only in this image to claim that "Re+1 preserves the underlying D3h lattice symmetry....while the STM topography of Re0 and Re-1 appear slightly distorted." (first paragraph on page 7).
- 5) It seems to me that a filled state at ~-0.4eV should be present on the diagram of Fig. 2d for Re0. Was it purposely not shown?
- 6) At the end of page 5, where it says "The calculated density of states (DOS) map of the frontier orbitals shown in Fig. 1i are in excellent agreement with the experimental differential conductance (dI/dV) maps". It would be beneficial to mention that the (dI/dV) maps are in fig. 1(e).
- 7) On page 9, a discussion of magnetic properties of the Re doped sample is carried out. However, the authors do not present any magnetic measurements in their sample and only rely on previous work. Do

the authors plan on characterizing the magnetic properties the samples using VSM/SQUID/MCD or any other experimental tool?

8) Still on page 9, in which figure can this statement be verified: “The unusually broad defect resonances on the order of 300meV observed for some Re0 and Re-1 states raises the question about the broadening mechanism”? Please make it clear by adding a reference to that figure. Same for the statement made on the top of page 10: “the measured defect state broadening does not decrease for Re in bilayer MoS2”.

9) The method used is well explained and appropriate for the purposed of achieving the expected outcomes. In particular, the results originated from STM, STS mapping and AFM also seem legit and reasonable. However, some STM images lack atomic resolution and are used as justification for symmetry comparison between the defect and the substrate, which is not ideal.

10) The difference between panels (a) and (b) showing dI/dV spectra on each of the figures S6, S7 and S8 are unclear. Were (a) and (b) taken in two different regions? That should be clarified. Is the black curve representing the substrate only? A legend is needed.

11) Some essential information are in the supplement material, which makes the text more challenging to follow, as one need to keep going back and forth between the two files. In particular, I believe that figure S2 and S4 and their related discussion should be part of the main manuscript.

12) The authors should add the current value used on each STM topography image. Apart from this, the experimental procedures provided in the methods section are sufficient for the experiment to be reproduced.

13) Could the authors please clarify why it is considered remarkable that “the orbital image of the broad defect state resonance undergoes a continuous evolution of contrast, depending on the energy at which the defect state is probed”. Isn’t it expected that the contrast in dI/dV mapping will depend on different sample voltages? What makes it an “unusual progression of the STS contrast within the defect resonance and its significant broadening”? Please forgive me if I am missing any piece of information here.

Response to Referee Report: NCOMMS-23-39170

We thank all three referees for their thorough and constructive feedback. In the following, we reply to each comment raised by the referees.

Reviewer: 1

Reviewer's comment. *Xiang and co-authors report the observation of charged defects on a transition metal dichalcogenide. They claim that they image a broken symmetry state by means of Scanning Tunneling Microscopy and Spectroscopy. While I believe that the experimental data are a beautiful characterization of the TMD defects and the experiments are carefully performed, I do not believe the current manuscript significantly contributes to the advancement of the field. I also believe that the interpretation of the data in terms of broken symmetries is wrong. Below I list my concerns.*

Authors' answer. We appreciate the referee's time in reviewing our manuscript and engaging in the scientific discussion. While acknowledging the referees skepticism about the interpretation of the symmetry breaking, it is important to note that this phenomenon has been extensively discussed and documented in existing literature, which we have referenced in our manuscript. Furthermore, our view is also shared by the other two referees. In the subsequent point-by-point discussion, we aim to address any possible ambiguities in the used terminology and highlight the significance of our work.

Reviewer's comment. *The main claim is the observation of symmetry breaking. However, what they observe is a defect with a particular orbital distribution. First, discrete symmetries are described in crystals. A defect is not the crystal, it does not follow the crystalline lattice – that's why it is a defect. One could argue that the defect breaks translational symmetry, but that does not mean much in terms of the underlying physics for the material itself (unless its properties are determined by disorder). Moreover, it wouldn't be a spontaneous symmetry breaking if it happens through lattice distortions (whether is a defect or strain).*

Authors' answer. There seems to be a misunderstanding regarding the term "symmetry breaking". We agree with the referee that any type of crystal defect breaks the translational symmetry by definition. However, in the paper we refer to the disruption of the local site symmetry by non-composite point defects in absence of any external strain or other imperfections. In general, any point defect such as vacancies, substitutions, interstitials are classified by their lattice site known as the Wyckoff position. The crystal field of the point

defect is expected to follow the point symmetry of this lattice site, which can be described by the point group. For instance, a chalcogen site defect (like the S vacancy) should follow a C_{3v} point symmetry and a transition metal site defect (like Re substituting for Mo) should follow a D_{3h} point symmetry. Deviation from this underlying point symmetry is commonly referred to as symmetry breaking [Drabold and Estreicher "Theory of Defects in Semiconductors", Springer (2007); Bersuker J. Phys.: Conf. Ser. 833 012001 (2017); Mosquera-Lois et al., npj Comp. Mater. 9, 25 (2023)].

In our paper, we demonstrate that Va_{CS}^- , Re_{Mo}^- , and Re_{Mo}^0 have an atomic and electronic structure that deviate from their expected point symmetry. The driving force behind this deviation is the (pseudo) Jahn-Teller effect (JTE). In fact, the Jahn-Teller theorem states, "A nonlinear polyatomic system in a spatially degenerate electronic state distorts spontaneously in such a way that the degeneracy is lifted and a new equilibrium structure of lower symmetry is attained." [Jahn & Teller, Proc. R. Soc. Lond. A 161, 220 (1937)].

The theory on Jahn-Teller systems that break the point symmetry of the lattice site is widely established and there is precedence for a number of such systems in bulk solids [cf. Goodenough, Annu. Rev. Mater. Sci. 27, 1 (1998)]. Nonetheless, experimental validation of such systems is non-trivial and has not been shown for 2D materials. Here, we directly visualize the Jahn-Teller distortion in both the atomic structure (Fig. 3) and electronic defect orbitals (Fig. 1 and Fig. 5). This experimental result is unambiguous and in excellent agreement with *ab initio* density functional theory.

Reviewer's comment. *As far as I understood, the observed two-fold LDOS on the defects comes from the orbital splitting. As a consequence, some orbitals are filled, some others are empty, and their distribution is two-fold instead of three-fold. That does not break the symmetry, it actually follows the symmetry of the orbitals (or a combination of them for that matter).*

Authors' answer. We try to clarify this point with a general comment and a specific example. On a fundamental quantum mechanical level, if the Hamiltonian \hat{H} of a system is invariant under a symmetry operation \hat{S} : $[\hat{H}, \hat{S}] = 0$, then any eigenvector $|\psi_e\rangle$ of \hat{S} is an eigenvector of \hat{H} . Consequently, any electronic defect orbital is invariant under the same symmetry operation as the underlying atomic structure.

Consider a symmetric sulfur vacancy in a MoS_2 lattice as expected for this lattice site. The defect's electron density at any given energy needs to be three-fold symmetric as well. In dI/dV orbital images at any given energy, all states will appear three-fold symmetric. This

is the case for the neutral S vacancy [cf. Schuler et al. Phys. Rev. Lett. 123, 076801 (2019)]. Observing a two-fold symmetric LDOS in this scenario is impossible. For the negative sulfur vacancy, an electron populates a doubly degenerate state (see Fig. 1f), which is a classic Jahn-Teller instability. This system can gain energy (116 meV) by distorting the lattice. Hence, the defect has now a lower symmetry than before $C_{3v} \rightarrow C_{2v}$ (referred to as symmetry breaking) and its associated orbitals appear two-fold symmetric as expected from the distorted lattice.

Reviewer's comment. *In general, I think the authors fail on clearly explaining what new physics we learn from their experimental findings, what the significance is.*

Authors' answer. In this paper, our objective is not to assert the discovery of novel physics, but rather to present, for the first time the occurrence of both a Jahn-Teller and a pseudo Jahn-Teller effect in two distinct defect systems within 2D materials through experimental observation and supported by theoretical modelling. Although such an effect has been predicted theoretically [Cheng et al. Phys. Rev. B 10, 100401 (2013); Komsa et al. Phys. Rev. B 12, 125304 (2015); Tan et al. Phys. Rev. Mater. 6, 064004 (2020)], the experimental verification has been lacking, especially in 2D materials. The direct observation of JTE in 2D materials is nontrivial and marks an important milestone.

By changing the substrate's chemical potential (or analogously an applied external gate voltage), we can manipulate the orbital symmetry of the defect and consequently the defect's spin and optical emission properties. This is of significant relevance for the 2D quantum emitter community.

We also refer to the two other referees who commented our work. Referee 2: "...these results are really interesting for a lot of people working on 2D materials." Referee 3: "This work is significant to the field, as it thoroughly characterizes doped MoS₂ and discusses the impact of different charge dopants and their symmetry on MoS₂ electronic properties. To the best of my knowledge and search, there are no systematic dI/dV mapping studies on doped MoS₂ that discuss charge state and symmetry breaking."

Reviewer's comment. *On a side note, the authors claim to do chemical gating and to 'control the Fermi level alignment through the substrate chemical potential'. How do the authors control that? It is phrased in a way that it seems like they gate the system, but I think that it's just the natural*

doping from the fact that the MoS₂ is on top of a substrate. If they don't control the doping with an external voltage through an insulating layer, they should not use the term 'gating'.

Authors' answer. We agree with the referee that the term "gating" is mainly used in the context of electrostatic gating by applying an external gate voltage. Here we refer to a change of the substrate chemical potential. Epitaxial graphene and quasi-freestanding epitaxial graphene are used as substrates with a very different work function, thereby substantially shifting the Fermi level in the TMD, akin to the impact of an external gate voltage. However, this potential is "built-in" and cannot be adjusted. This approach is related to alkali metal adsorption commonly employed for in-situ *n*-doping of 2D materials in angle-resolved photoemission spectroscopy [e.g. Zhang, Y. et al. Nano Lett. 16, 2485–2491 (2016)].

Action taken. To avoid any misunderstandings we replaced the term "chemical gating" by "varying the substrate chemical potential" throughout the manuscript.

Reviewer: 2

Reviewer's comment. *In their article "Charge State-Dependent Symmetry Breaking of Atomic Defects in Transition Metal Dichalcogenides", Feifei Xiang and coworkers combine state-of-the-art scanning probe experiments with DFT calculations to investigate 2 types of defects in CVD-grown monolayer MoS₂. For the quite common S vacancies the authors find two different configurations once the defect level is charged forming VacS⁻¹ confirming a prediction which was made in Ref. 22 of the manuscript. The authors then continue to investigate Re dopants as those can be prepared in different charged states in contrast to the VacS. And indeed, also for the ReMo-1 the authors find a broken-symmetry ground state. While the symmetry-broken S vacancy is due to a "normal" Jahn-Teller distortion, the authors find that the symmetry reduction for the Re dopants is due to a pseudo Jahn-Teller effect.*

I find the manuscript well written and the results convincing. Furthermore, I think that these results are really interesting for a lot of people working on 2D materials and it could be published. Yet, the distortions found in MoS₂ might be also specific to this system and thus, in order to justify better the publication in Nat. Commun., I think the authors should add a paragraph at the end, if the distortions can also be expected in other systems such as the other semiconducting 2H-TMDCs or -even better - check for, e.g., WSe₂.

Otherwise I have only a few minor points concerning the calculations which however also need to be addressed before any publication.

Authors' answer. We thank the reviewer for their positive feedback and constructive suggestions/comments. The direct observation of Jahn-Teller and pseudo-Jahn-Teller effects is important for far more than just the systems examined in this work. Jahn-Teller and pseudo-Jahn-Teller effects are predicted in a wide variety of transition metal compounds. Indeed, we theoretically predict the same Jahn-Teller distortions also for the -1 charge state of the S vacancy in WS₂ and Se vacancy in WSe₂, despite the much stronger SOC, as shown in Figure R1.

Action taken. We added a short discussion about the prediction of the same symmetry breaking also in WS₂ and WSe₂ at the end of the paragraph "Jahn-Teller Driven Symmetry-Breaking of VacS⁻¹ in MoS₂". We also added Figure R1 to the SI.

Figure R1: Jahn-Teller effect for the negatively charged S vacancy in WS₂ and Se vacancy in WSe₂. (a-c) Projected density of states of Vac_S in WS₂ in the +1, 0, and -1 charge state. Vac_S⁻¹ exhibits a symmetry-broken HOMO, shown in (e) while in the positive and neutral charge state the C_{3v} symmetry is preserved. (f-h) Projected density of states of Vac_{Se} in WSe₂ in the +1, 0, and -1 charge state. Vac_{Se}⁻¹ exhibits a symmetry-broken HOMO, shown in (j) while in the positive and neutral charge state the C_{3v} symmetry is preserved.

Reviewer's comment. at the end of page 5, the authors write (when speaking about the possibility of having VacS-2: "Possibly, the system is trapped in a local minimum with a potential barrier too high to relax to the global potential energy minimum."

This could be checked maybe via CI-NEB, or at least a mapping as shown in Fig. 4 for the Re dopant. Especially checking along a linear interpolation of the geometries of the symmetric/distorted structures.

Authors' answer. We thank the reviewer for this suggestion. Initially we proposed two possible reasons for the observation of symmetric vacancies (in addition to the expected distorted ones): (i) the -1e vacancies could be trapped in a local minimum at the symmetric position, or (ii) the vacancy is -2e charged (not -1e) where a symmetric ground state is expected [Tan et al. Phys. Rev. Materials 4, 064004 (2020)].

As suggested by the referee, we performed additional calculation of the Vac_S^{-1} potential energy change ΔE along a linear interpolation between the distorted and symmetric geometries, as shown in Figure R2. We find indeed no indications of a local potential minimum. Therefore, we removed this claim from the paper.

Figure R2: Linear interpolation of PES between the distorted ground state and symmetric configuration of Vac_S^{-1} in MoS_2 .

Action taken. We revised the paragraph on the discussion of the symmetric S vacancies, removing the claim about the possibility of a local potential minimum at the symmetric configuration.

"Possibly, the symmetric variant is in fact a doubly negatively charged vacancy Vac_S^{-2} , which is not Jahn-Teller active [Tan et al. Phys. Rev. Materials 4, 064004 (2020)]. Importantly, these findings are not limited to MoS_2 , but similarly apply to the negatively charged S vacancy

in WS₂ and negatively charged Se vacancy in WSe₂ (see Fig. S15), despite their significantly stronger SOC."

Reviewer's comment. *Why did the authors not include SOC in their calculations? It is known to be important in TMDCs and also the defect states in WS₂ and WSe₂ split in two separate bands due to SOC as shown in, e.g., Ref. 34 and <https://doi.org/10.1038/s41699-023-00421-0>, respectively. Even if the Mo-based TMDCs have much lower SOC splitting, I think the authors should also check if the inclusion of SOC changes some of their conclusion.*

Authors' answer. We thank the referee for this question. We agree that SOC is an important interaction in TMDs, which is also true for the chalcogen defect state as correctly pointed out by the referee. Indeed, we did check that the inclusion of SOC does not change the observation of defect level splitting/symmetry breaking. In the Figure R3 below, we show the projected density of states of the Vac_S in MoS₂ for the neutral and negative charge state. As can be seen in Figure R3 b, SOC slightly lifts the degeneracy of the Vac_S⁰ defect state. However, the splitting is rather small, as commonly observed for Mo 4*d* states as also pointed out by the referee. In the negative charge case, we observe the JT distortion with a very similar density of states with and without SOC. As mentioned in reply to the first question, also for W compounds we see qualitatively the same behavior despite the stronger SOC coupling of W 5*d* states.

For simplifying the discussion and save computational costs we therefore decided not to include SOC in the Re substitution case. Especially for the adiabatic potential energy surface, including SOC would be prohibitively long. Moreover, while SOC can strongly impact JT behavior, it will not qualitatively alter the effect for this *d*-orbital occupation and D_{3h} point group [cf. Phys. Rev. X 10, 031043 (2020)].

Action taken. We added the following paragraphs to the SI:

"In Figure R3 the projected density of states of Vac_S in MoS₂ in the neutral and negative charge state with and without considering SOC is shown. As can be seen in Figure R3b, SOC slightly lifts the degeneracy of the Vac_S⁰ defect state. However, the splitting is rather small, as commonly observed for Mo 4*d* states. In the negative charge state, we observe the JT distortion with a very similar density of states with and without SOC, as seen in Figure R3c and Figure R3d, respectively. Therefore we can conclude that while effects of SOC are not quantitatively negligible, the system exhibits the same qualitative behavior with and without SOC.

Moreover, the observation of the symmetry-broken ground state of the chalcogen vacancy is not exclusive for the S vacancy in MoS₂ but is also predicted for the negatively charged S vacancy in WS₂ and negatively charged Se in WSe₂, as shown in Figure R1. Despite the much stronger SOC for W 5d states, also in this case the effect of SOC is negligible for the qualitative observation of the Jahn-Teller effect."

Figure R3: Comparison of Vac_S^0 and Vac_S^{-1} in MoS₂ with and without considering SOC. While SOC has an effect on the defect state splitting of the neutral Vac_S , the Jahn-Teller splitting is observed for Vac_S^{-1} with or without considering SOC.

"In Figure R4 the projected density of states of Re_{Mo} in MoS₂ in the neutral and negative charge state with and without considering SOC is shown. In both cases, the effect of SOC is negligible. In particular, the pseudo-JT distortion is reproduced."

Figure R4: Comparison of Re_{Mo}^0 and Re_{Mo}^{-1} in MoS_2 with and without considering SOC. The effect of SOC is in both cases negligible.

Reviewer's comment. A 4x4 supercell seems to be rather small, especially for a charged defect. In similar calculations for uncharged chalcogen vacancies, I found that there was still a considerable interaction between the repeated defects. The authors should check that their results do not depend on an increase to 5x5 or need to discuss the possible interaction at least.

Authors' answer. Previous work [Tan et al. Phys. Rev. Materials 4, 064004 (2020)] showed the convergence as a function of supercell size, where a 4x4 cell has been found sufficient to avoid significant electronic overlap of defect orbitals between super cells. However, to improve the k-point density consistently across systems, a 5x5 supercell has been used instead. For consistency, we repeated all calculations using a 5x5 unit cell.

Action taken. The 4x4 supercell on the S vacancy system is replaced with a 5x5 supercell in all results.

Reviewer's comment. *The k-point grids are only given for the DOS calculations. The grids for the relaxation is needed as well. Why 6x6x2 for a 2D system? (why 2 in z direction) Also, why did the authors use the tetrahedron method in one case but a Gaussian smearing in the other? Also, why the different functionals for the different defects? I find it surprising that for the 2 defects so many parameters were changed – also the k-point density (for the DOS calculations) are different.*

Authors' answer. We agree with the referee that the computational parameters should be more consistent, and we should also include the relaxation k-point grids. The Rhenium substitutional defect data in the symmetric and symmetry-broken structures has been recalculated with the same computational parameters used to calculate the sulfur vacancy systems. The 6x6x2 grid was not needed for the 2D system, and shows no difference in the result compared to a 6x6x1 grid.

Action taken. In the revised version, we considerably improved the consistency of the calculation. Now in all calculations aside from the adiabatic potential energy surfaces, we used a 5x5x1 Γ -centered Monkhorst-Pack k-point mesh. We also treated the exchange-correlation in calculations aside from the potential energy surface using the strongly constrained and appropriately normed meta-generalized gradient approximation functional with van der Waals interactions (SCAN+rVV10). In the calculations used to create the adiabatic potential energy surfaces, the computational cost was reduced by using a Γ point-only calculation in the PBE functional.

We updated Fig. 1f,g, Fig. 4d-g, and all theory figures in the Supplement using these consistent set of parameters.

Reviewer: 3

Reviewer's comment.

Noteworthy results:

- *Impact of substrate choice on the Fermi level placement of MoS₂ and charged states of Re dopants.*
- *Systematic STS study on sulfur vacancies and Re dopants on MoS₂ providing insight on Re multiple stable charge states and sulfur vacancies symmetry.*

Relevance:

- *This work is significant to the field, as it thoroughly characterizes doped MoS₂ and discusses the impact of different charge dopants and their symmetry on MoS₂ electronic properties. To the best of my knowledge and search, there are no systematic dI/dV mapping studies on doped MoS₂ that discuss charge state and symmetry breaking. This work corroborates to the advance of knowledge that impact tunability of optical emission properties and spin-photon interfaces for future electronic applications. The manuscript is well-written, and the topic relevant. However, there are some points to be discussed and clarified before I can make the recommendation for this manuscript to be published fitting the high standards of Nature Communications journal:*

Authors' answer. We highly appreciate the detailed and constructive feedback of the referee and the positive assessment of our work. In the following, we address the referee's points to clarify our work and highlight changes in the revised version based on these suggestions.

Reviewer's comment. *How were the intrinsic defects identified? The authors claim that OS, CHS, CrMo and WMo are also present in the sample, apart from the studies VacS, but no convincing method was described as to how each type of intrinsic defect was properly identified. Are there simulated STM images that support these claims? Otherwise, the somehow arbitrary identification of defects is not convincing.*

Authors' answer. This is a valid point and there has been indeed many false assignments of TMD point defects in the early literature based on STM imaging alone. We base our assignments on a series of our previous studies of TMD point defects [Barja et al., Nat. Commun., 10, 3382, (2019); Schuler et al., ACS Nano, 13, 10520 (2019); Schuler et al., Phys. Rev. Lett., 123, 076801 (2019); Cochrane et al., 2D Mater., 7, 031003 (2020); Kozhakhmetov et al., Adv. Mater., 32, 2005159 (2020); Mitterreiter et al., Nano Lett., 20, 4437 (2020); Kozhakhmetov et al., Adv. Funct. Mater., 2105252 (2021); Torsi et al. ACS Nano 17, 15629 (2023)]. In these papers, we rigorously characterize the defect type using a combination of techniques, including STS of in-gap defect states, STS orbital imaging, CO-tip nc-AFM

atomic contrast, and *ab initio* modeling. Most relevant for this publication we previously identified sulfur vacancies in WS₂ [Schuler *et al.*, Phys. Rev. Lett., 123, 076801 (2019)] and MoS₂ [Mitterreiter *et al.*, Nano Lett., 20, 4437 (2020)], and Re dopants in WSe₂ [Kozhakhmetov *et al.*, Adv. Mater., 32, 2005159 (2020)] and MoS₂ [R. Torsi *et al.* ACS Nano 17, 15629 (2023)]. These specific types of defects can be deliberately generated by UVH annealing (S vacancies) and chemical doping (Re substitutions), respectively. Therefore, we can identify these type of atomic point defects with certainty.

Regarding other minority defects that may be present in as-grown samples (without high annealing or intentional doping), such as O_S, CH_S, Cr_{Mo} and W_{Mo}, we assign them based on their comparable STM contrast observed in other TMDs when imaged close to the conduction band onset, in conjunction with their STS fingerprint. In Figure R5 below, we present STM images of point defects in different TMD materials. Notably, STM images obtained at 100-200 mV above the TMD conduction band edge allow us to confidently assign the type of defect, leveraging our prior knowledge acquired from studies in other TMD semiconductors. This consistent contrast is observed in both multilayers and bulk TMDs. However, for the scope of this paper, the assignment of these other defects is not within the primary focus.

[REDACTED]

Figure R5: STM topography of common as grown point defect in different TMD materials, recorded close to the conduction band onset. The characteristic contrast is comparable for different TMDs, as well as mono- and multilayers. A rigorous assignment has been performed in the cited papers and is mainly based on their STS fingerprint and orbital imaging (as shown on the right side for Cr_W).

Action taken. In Figure S2, we have provided explicit clarification in the caption that the defect assignment relies on prior work: "...monolayer MoS₂/QFEG along with other as-grown defects such as O_S [12,13], CH_S [14], V_{Mo} [15], Cr_{Mo} [13], and W_{Mo}, labeled in black. The defect assignment is grounded in comprehensive characterization outlined in the referenced works. The STM contrast 100-200 mV above the TMD conduction band edge in conjunction with their unique STS fingerprint allow us to reliably assign defect types across various TMD semiconductors. ... "

Reviewer's comment. *On the Re doping study, again a better argument is needed to guarantee that those were Re substitutionals. How can the authors be confident that they are not defects from*

other nature, such as the ones previously cited. Simulated STM images would again help with this discussion point.

Authors' answer. Again we appreciate the referee's request for a rigorous defect assignment. Achieving an unambiguous defect assignment can be accomplished through two distinct approaches: (i) a comparative analysis of experimental results and theoretical modeling, or (ii) a targeted synthesis followed by comparative experiments, involving samples with and without intentionally introduced defects. Here we did both: In Fig. 5b and Fig. 4g the experimental STS maps of Re_{Mo}^0 and calculated LDOS yield a good agreement. In Fig. S14 we provide additional simulated LDOS maps of Re in different charge states and in Fig. S6-S8 experimental STS maps. Moreover, in our previous study [Torsi et al. ACS Nano 17, 15629 (2023)], we describe in detail the density controlled introduction of Re substitutional dopants in MoS_2 by MOCVD. The successful incorporation of these dopants is confirmed by Z-contrast scanning transmission electron microscopy (STEM), inductively coupled plasma mass spectrometry (ICP-MS), and x-ray photoelectron spectroscopy (XPS), as shown in Figure R6a-e below. In the current study, we compared MoS_2 samples with and without intentional Re-doping as shown in Figure R6f. While the pristine MoS_2 sample shows no Re dopants (as expected), the 0.5% Re-doped sample shows many Re dopants together with other 'native' defects (at much lower concentration) that are also present in the pristine sample. Due to this comparative study and the quantitative match of the expected Re concentration in doped-samples we can confidently identify Re dopants.

[REDACTED]

Figure R6: Re-doped MoS₂. (a) Density controlled Re doping of MoS₂ in a MOCVD process. (b) Linear relation of the Re density in the MoS₂ as a function of the Re₂CO₁₀ precursor flow. (c,d) Z-contrast scanning transmission electron microscopy (STEM) of Re substitutional dopants in Re-doped MoS₂. (e) X-ray photoelectron spectroscopy (XPS) verifying successful Re incorporation. Panels (a-e) are adapted from [Torsi et al. ACS Nano 17, 15629 (2023)]. (f) STM topography of a pristine MoS₂ sample (left) with no Re dopants and a 0.5% Re-doped MoS₂ sample with many Re dopants (right).

Reviewer's comment. *The authors mentioned in page 6 that "Counter-intuitive for an n-type dopant, this results in the Re impurities becoming negatively charged, i.e. they accept an additional electron.". Since it is counter-intuitive, could the authors present a better explanation for why the charge transfer is still happening. Forgive me if the explained that somewhere else if I missed it. It would still be beneficial to include this discussion directly following the statement.*

Authors' answer. We thank the referee for the feedback. The presence of a negative charge on an *n*-type dopant may indeed seem counter-intuitive, especially when considering a simplified model with only a single dopant state in mind. However, this particular dopant features multiple states closely spaced in energy. Here we control the occupation of defect levels by the substrate work function, which pins the Fermi level within the MoS₂ band gap. On epitaxial graphene (EG) substrates, the Fermi level is positioned significantly higher in the band gap, only ~200 meV below the MoS₂ conduction band onset. This configuration allows a formally unoccupied Re state to be populated, resulting in a negative charge on the dopant. Conversely, with quasi-freestanding epitaxial graphene (QFEG) substrates, the Fermi level is situated much lower,

approximately 700 meV below the edge of the MoS₂ conduction band. Consequently, Re dopants are typically found in either their neutral or positive charge state.

The different Fermi level alignment can be seen in the STS measurements shown in Fig. 2e, where the black curve corresponds to MoS₂/QFEG and the gray curve represents MoS₂/EG. A corresponding schematic level diagram is provided in Fig. 2d.

Action taken. To clarify that the much smaller work function of EG leads to a Fermi energy very close to the conduction band edge of MoS₂, and correspondingly to the occupation of a formally unoccupied Re state, we modified the text as follows: "By using epitaxial graphene (EG) without hydrogen intercalation as a substrate, which has a substantially smaller work function than QFEG, we push the Fermi level even higher. [41,42] **As a consequence**, this results in the Re impurities becoming negatively charged, i.e. they accept an **additional** electron, **somewhat counter-intuitive for an n-type dopant.**"

Reviewer's comment. *Since the STM topography image on fig. 3(c) presents a very strong contrast difference and lacks atomic resolution, it is an overstatement to rely only in this image to claim that "Re+1 Mo preserves the underlying D_{3h} lattice symmetry....while the STM topography of Re⁰ Mo and Re-1 Mo appear slightly distorted." (first paragraph on page 7).*

Authors' answer. We agree with the referee's statement. However, our assignment of the three-fold symmetry of Re⁺¹_{Mo} is not based on the STM image in Fig. 3(c) alone (as the text may suggest) but relies on all Re⁺¹_{Mo} defect orbitals being three-fold symmetric as clearly shown in Fig. S6. This is a conclusive sign that the defect preserves the underlying lattice symmetry.

Action taken. We adjusted the text in the following way: "Most importantly, Re⁺¹_{Mo} preserves the underlying D_{3h} lattice symmetry, as previously reported for Re⁺¹_W in WSe₂, [39] while the STM topography of Re⁰_{Mo} and Re⁻¹_{Mo} appear slightly distorted. **This is consistent with their imaged defect orbital symmetries discussed next.**"

Reviewer's comment. *It seems to me that a filled state at ~-0.4eV should be present on the diagram of Fig. 2d for ReMoO. Was it purposely not shown?*

Authors' answer. We appreciate the keen observation of the referee. The referee is correct, that we did not show two individual defect states for the singly occupied orbital

of Re_{Mo}^0 in the schematic level diagram. Due to the Coulomb energy associated with adding or removing an electron to or from the same orbital, a singly occupied orbital will appear both in the occupied and unoccupied state [Cochrane et al. Nat. Commun. 12, 7287 (2021)]. This is commonly observed in singly occupied molecular orbitals (SOMO/SUMO), which can be assigned as such by orbital imaging [e.g. Repp et al. Science, 312, 1196 (2006)]. Since the simplified one-particle level diagram does not account for this effect, we just show a single level with half occupation for simplicity. To avoid any confusion we clarified the text.

Action taken. We clarify in the text that for a singly occupied defect orbital we expect to see two STS resonances: "Based on the STS spectra we derive a simplified schematic level diagram for the differently charged Re_{Mo} impurities in Fig. 2d. **Note that for Re_{Mo}^0 we only draw one level at the Fermi energy despite two STS resonances being observed around the Fermi level. Due to the Coulomb energy associated with adding or removing an electron to or from a singly occupied orbital, we observe two resonances both in the occupied and unoccupied state [Repp et al. Science, 312, 1196 (2006), Cochrane et al. Nat. Commun. 12, 7287 (2021)].**"

Reviewer's comment. *At the end of page 5, where it says "The calculated density of states (DOS) map of the frontier orbitals shown in Fig. 1i are in excellent agreement with the experimental differential conductance (dI/dV) maps". It would be beneficial to mention that the (dI/dV) maps are in fig. 1(e).*

Authors' answer. We thank the referee for this comment. We added the reference to the experimental dI/dV maps.

Action taken. Added to the text: "...are in excellent agreement with the experimental differential conductance (dI/dV) maps **in Fig. 1e**"

Reviewer's comment. *On page 9, a discussion of magnetic properties of the Re doped sample is carried out. However, the authors do not present any magnetic measurements in their sample and only rely on previous work. Do the authors plan on characterizing the magnetic properties the samples using VSM/SQUID/MCD or any other experimental tool?*

Authors' answer. We thank the referee for this valuable suggestion. Indeed the magnetic properties of the Re dopants and charged vacancies are highly interesting. At present, we

do not have the capabilities in-house to perform any of the proposed measurements. Given the low probe volume, we anticipate that some of the proposed measurements may also be challenging even at high defect concentrations. We are currently working with collaborators at ETH Zurich to perform STM electron spin resonance (ESR) that has a single atom sensitivity [Baumann et al. Science 350, 417 (2015)]. However, this is work in progress and beyond the scope of the manuscript.

Reviewer's comment. *Still on page 9, in which figure can this statement be verified: "The unusually broad defect resonances on the order of 300meV observed for some $Re^{+0}Mo$ and $Re^{-1}Mo$ states raises the question about the broadening mechanism"? Please make it clear by adding a reference to that figure. Same for the statement made on the top of page 10: "the measured defect state broadening does not decrease for Re in bilayer MoS₂".*

Authors' answer. We agree that the figure reference may not be obvious at this point in the text. The unusually broad resonances were initially discussed on page 7, where we mentioned, "The pink dI/dV spectrum in Fig. 2e and defect orbitals depicted in Fig. S6 for Re_{Mo}^{+1} closely resemble the Re_w^{+1} states observed in WSe [2,39] with several defect states just above the Fermi level. **Re_{Mo}^0 (Fig. 2e blue, and Fig. S6)** exhibits several unoccupied defect states above Fermi, along with an **unusually broad defect resonance below it**". To provide further context, on page 9, we again reference to the spectrum for comparison.

Action taken. We added Figure references in the text:

"The unusually broad defect resonances on the order of 300 meV observed for some Re_{Mo}^0 and Re_{Mo}^{-1} states (Fig. 2e blue and green) raises the question about the broadening mechanism."

", and the measured defect state broadening does not decrease for Re in bilayer MoS₂ (Fig. 5a).

Reviewer's comment. *The method used is well explained and appropriate for the purposed of achieving the expected outcomes. In particular, the results originated from STM, STS mapping and AFM also seem legit and reasonable. However, some STM images lack atomic resolution and are used as justification for symmetry comparison between the defect and the substrate, which is not ideal.*

Authors' answer. We thank the referee for appreciating the approach and quality of our experimental methods. As the referee is probably well aware, the STM contrast is highly

sensitive on the applied bias since it probes the electronic density of states. In our study, most STM topographies were recorded within a few 100 meV above the conduction band onset where the defects can most easily be distinguished and compared to other materials. While atomic resolution is indeed easily achievable on this material by choosing a bias value within the band gap, it is important to note that its significance is rather limited given there is a strong tip dependence of the atomic STM contrast. Here we use CO-tip nc-AFM to resolve the atomic lattice, which is a much better indicator for the lattice registry of the defect and the local atomic distortions than STM. With STS we have a powerful tool to probe the electronic defect orbitals, where the electronic orbital density is inherently distributed over several lattice sites. In this way, we observe the drastic electronic modification in response to the broken defect symmetry. Hence, we firmly believe that atomically-resolved nc-AFM and orbital imaging by STS offers a more robust and insightful means of investigating both the atomic and electronic properties of defects as compared to atomically-resolved STM.

Reviewer's comment. *The difference between panels (a) and (b) showing dI/dV spectra on each of the figures S6, S7 and S8 are unclear. Were (a) and (b) taken in two different regions? That should be clarified. Is the black curve representing the substrate only? A legend is needed.*

Authors' answer. Both colored dI/dV spectra in panels (a) and (b) were indeed acquired in the same spatial location. Panel (b) corresponds to data collected at a reduced tip-sample distance and within a narrower bias window, facilitating a more precise resolution of the frontier defect states. As correctly noted, the gray spectrum represents measurements taken on the pristine substrate in close proximity to the defect, serving as a reference.

Action taken. We added a legend to each Figure and clarified in the caption that the STS spectra in (b) are recorded at smaller tip-sample distances: "**(a,b)** dI/dV spectra of $\text{Re}_{\text{Mo}}^{+1}$ (pink) with the main defect resonances indicated, and reference spectrum on pristine MoS_2 in gray. In panel b, data is acquired at a reduced tip-sample distance, emphasizing frontier defect states within a smaller bias window."

Reviewer's comment. *Some essential information are in the supplement material, which makes the text more challenging to follow, as one need to keep going back and forth between the two files. In particular, I believe that figure S2 and S4 and their related discussion should be part of the main manuscript.*

Authors' answer. We thank the referee for this suggestion and we agree that the extended data figures in the Supplemental Material provide relevant information for the reader. While we acknowledge the importance of Figures S2 and S4 in providing additional context, we are constrained by the space limitations in the main manuscript, which already accommodates five full-page width figures. In the main text, we refer to supplementary figures to indicate the availability of further material for interested readers. However, none of the supplementary figures is essential to understand the main conclusions of the paper.

In Fig. S2 we show other point defects in the MOCVD-grown MoS₂ recorded at different biases, which is not essential for the story on deliberately introduced S vacancies and Re dopants. Similarly, Fig. S4 presents additional STM images and dI/dV maps of symmetric and distorted S vacancies in the top and bottom layer. We refer the referee to Fig. 1a-e, which shows essentially the same type of data. Notably, some of the images are exact duplicates (Fig. 1a = Fig. S4b; Fig. 1b = Fig. S4c; Fig. 1c = Fig. S4d; Fig. 1e top row = contained in Fig. S4h). These panels were replicated in Figure S4 to facilitate a direct comparison with the symmetric S vacancies, which are not featured in Fig. 1. This is also indicated in the caption of Fig. S4: "Panels b-d, and h are the same as in Fig. 1a-c, and Fig. 1e, but are reprinted for comparison." Given these considerations, we believe that the inclusion of Figure S2 or Figure S4 in the main text would offer limited additional value.

Reviewer's comment. *The authors should add the current value used on each STM topography image. Apart from this, the experimental procedures provided in the methods section are sufficient for the experiment to be reproduced.*

Authors' answer. We agree with the referee and took according actions.

Action taken. We added the current set point to all STM topography images.

Reviewer's comment. *Could the authors please clarify why it is considered remarkable that "the orbital image of the broad defect state resonance undergoes a continuous evolution of contrast, depending on the energy at which the defect state is probed". Isn't it expected that the contrast in dI/dV mapping will depend on different sample voltages? What makes it an "unusual progression of the STS contrast within the defect resonance and its significant broadening"? Please forgive me if I am missing any piece of information here.*

Authors' answer. We appreciate the referee's valid questions and welcome the opportunity to provide further clarification. In dI/dV spectroscopy, it is generally anticipated that distinct resonances in the dI/dV spectrum correspond to specific orbitals, each with its characteristic symmetry [Repp et al. Phys. Rev. Lett. 94, 026803 (2005)]. Consequently, mapping out different orbitals associated with different resonances is expected to yield different orbital shapes. However, it is not typically anticipated that probing the same defect orbital at slightly different energies would yield observable changes in contrast.

Strong electron-phonon interactions are a common mechanism resulting in orbital broadening. Nevertheless, dI/dV images of the same vibronically broadened orbital typically do not exhibit significant variations. Exceptions arise in cases where inelastically excited phonon modes influence the tunneling probability [Krane et al. Phys. Rev. Lett 124, 116804 (2020)] or when a functionalized STM tip is employed [Gross et al. Phys. Rev. Lett. 107, 086101 (2011), Pavlicek et al. Phys. Rev. Lett. 110, 136101 (2013)]. In our extensive study of TMD defect systems, we have not encountered such behavior so far. Typically, defect resonances exhibit much narrower linewidths (on the order of a few meV) [e.g. Schuler et al. Phys. Rev. Lett. 123, 076801 (2019)], or in cases of vibronically broadened states, individual vibronic peaks can be resolved [Cochrane et al. Nat. Commun. 12, 7287 (2021)]. Nevertheless, even in these instances, the shape of the orbital remains invariant with respect to bias. To the best of our knowledge, the observed continuous evolution of orbital contrast on a broadened, featureless resonance is unprecedented.

REVIEWERS' COMMENTS

Reviewer #1 (Remarks to the Author):

I appreciate the author's further explanation on the concept of symmetry breaking. If I am not mistaken, some of the observed defects undergo a distortion due to Jahn Teller effects which in turn break the symmetry, which is observed through STS in the orbital shape.

Regarding the gating issue, the authors changed the text to 'by varying the substrate chemical potential'. I still think this is problematic because it gives the impression the authors can actually vary the substrate chemical potential. And, unless I misunderstood, they do not. They mention it is built-in and cannot be adjusted. This should be clarified throughout the text, otherwise it would be an overstatement.

While I believe the authors when they say it's the first time this effect has been observed, I am sorry but I still question the relevance of such observation. I believe the author's interpretation though.

Reviewer #2 (Remarks to the Author):

I thank the authors for their extensive reply especially to the interesting experimental questions of referee 3. I can recommend publication since all questions were answered and additional calculations/analysis have been done.

Reviewer #3 (Remarks to the Author):

I think the authors addressed the raised concerns and modified the manuscript appropriately, specially when it comes to proper identification of defects by referring back to relevant references. They have also demonstrated deep knowledge in the process and were convincing about the accuracy of the results.

I feel confident in recommending this work for publication.

Response to Referee Report: NCOMMS-23-39170

We thank all three referees for reviewing our paper and recommending publication.

Reviewer: 1

Reviewer's comment. *I appreciate the author's further explanation on the concept of symmetry breaking. If I am not mistaken, some of the observed defects undergo a distortion due to Jahn-Teller effects which in turn break the symmetry, which is observed through STS in the orbital shape. Regarding the gating issue, the authors changed the text to 'by varying the substrate chemical potential'. I still think this is problematic because it gives the impression the authors can actually vary the substrate chemical potential. And, unless I misunderstood, they do not. They mention it is built-in and cannot be adjusted. This should be clarified throughout the text, otherwise it would be an overstatement. While I believe the authors when they say it's the first time this effect has been observed, I am sorry but I still question the relevance of such observation. I believe the author's interpretation though.*

Authors' answer. We appreciate that the referee changed their opinion based on the feedback provided.

To indicate that the chemical potential is indeed built-in and cannot be dynamically changed we revised the abstract and introduction like specified below.

Action taken. Revised the text:

Abstract: "By ~~varying~~ **changing** the **built-in** substrate chemical potential, different charge states of sulfur vacancies..."

Introduction: "Charge state tristability of Re dopants is achieved by varying the ~~substrate~~ chemical potential **via a different work function of the substrate.**"